# Genetic and Morphological Identification of *Spirometra decipiens* in Snakes and Domestic Dog Found in Cuba

**DOI:** 10.3390/pathogens11121468

**Published:** 2022-12-05

**Authors:** Alexander Morales, Rebeca M. Laird-Pérez, Virginia Capó, Enrique Iglesias, Luis Fonte, Arturo Plascencia-Hernández, Enrique J. Calderón, Keeseon S. Eom, Yaxsier de Armas, Héctor R. Pérez-Gómez

**Affiliations:** 1Pathology Department, Hospital Center, Institute of Tropical Medicine “Pedro Kourí”, Havana 11400, Cuba; 2Teaching Department, Institute of Tropical Medicine “Pedro Kourí”, Havana 11400, Cuba; 3Centro de Ingeniería Genética y Biotecnología, Habana 11400, Cuba; 4Parasitology Department, Institute of Tropical Medicine “Pedro Kourí”, Havana 11400, Cuba; 5Centro Universitario de Ciencias de la Salud, Universidad de Guadalajara, Guadalajara 44100, Mexico; 6Instituto de Biomedicina de Sevilla, Hospital Universitario Virgen del Rocío/Consejo Superior de Investigaciones Científicas/Universidad de Sevilla, 41013 Seville, Spain; 7Centro de Investigación Biomédica en Red de Epidemiología y Salud Pública (CIBERESP), 28029 Madrid, Spain; 8Department of Parasitology and Medical Research Institute, School of Medicine, Chungbuk National University, Chungbuk 361-763, Cheongju 28644, Republic of Korea; 9Department of Clinical Microbiology Diagnosis, Hospital Center, Institute of Tropical Medicine “Pedro Kourí”, Havana 11400, Cuba

**Keywords:** *Spirometra*, snakes, dog, molecular detection, morphology, *Spirometra decipiens*, Cuba

## Abstract

*Spirometra* (Cestoda: Diphyllobothriidea) affects humans and some species of domestic and wild animals which eventually interact with humans. In this article, we report three new cases of *Spirometra decipiens* (Diesing, 1850) infection observed in two intermediate hosts and one definitive host, in Cuba. Genetic and morphological identification of *S. decipiens* in two snakes and a domestic dog were carried out by molecular means and routine histological study using hematoxylin–eosin staining, respectively. Taken together, the anatomical location, the host species infected with the specimens and their morphological and genetic features, all the samples were identified as *S. decipiens*. In each of the three cases, PCR assays using specific primers amplified bands that corresponded to *S. decipiens* species. To our knowledge, this paper is the first report of *S. decipiens* in species of Cuban endemic fauna and in the Caribbean islands. These species constitute a real or potential risk of transmission of *Spirometra* to humans in Cuba.

## 1. Introduction

*Spirometra* Faust, Campbell and Kellogg, 1929 (Cestoda: Diphyllobothriidea) affects humans and some species of domestic and wild animals which eventually interact with humans. Additionally, infected secondary intermediate hosts can be ingested by paratenic hosts such as nonhuman primates, pigs *Sus scrofa* Linnaeus, 1758, rodents and birds that contribute to the long-term survival of the pathogen in the environment. Although human infection with adult worms is not common, infection of intermediate and incidental hosts usually results in plerocercoids migration through the tissues producing clinical manifestations [1]. 

Three intermediate hosts are involved in the life cycle of *Spirometra* spp. Copepods (Cyclopidea) may carry the procercoid larvae and they are ingested by secondary intermediate hosts such as: amphibians, reptiles, *Gallus gallus domesticus* Linnaeus, 1758 and pigs in which the plerocercoid stage occurs. Domestic dogs *Canis lupus familiaris* Linneaus, 1758 and domestic cats *Felis catus* Linaneus, 1758 are usually definitive hosts [2]. Two to three weeks later, adult tapeworms are formed and are reported to survive up to 30 years [3]. 

The complexity of the *Spirometra* spp. biological cycle is one of the reasons why the prevalence of infection in the definitive, as well as in the intermediate hosts, is very low in some areas (e.g., in South America 1.5%). However, in the Asian region, there are endemic areas where the prevalence in domestic cats, dogs and frogs is 40.5%, 27.5% and 51.9%, respectively. Other studies in Serbia have reported up to 57% prevalence in pigs and almost 100% infection of the grass snake *Natrix natrix* in Russia [4,5]. 

The classification of the *Spirometra* species is still confusing. The current phylogenetic analysis indicates that there are at least six different molecular well-defined lineages corresponding to five species. Most European specimens are *Spirometra erinaceieuropaei* (Rudolphi, 1819) [6]. *Spirometra folium* (Diesing, 1850) was described in samples from Africa mainly in Sudan, Ethiopia and Tanzania. [7]. *Spirometra mansoni* (Cobbold, 1882) was reported in specimens from Australia and Romania, a few specimens from Korea and Japan, and a single sequence from Tanzania [8,9]. Finally, *Spirometra decipiens* (Diesing, 1850) [10,11] and *Sparganum proliferum* (Ijima, 1905) [12,13] were identified in specimens from North and South America. 

The implementation of molecular biology techniques for pathogens identification became a very useful tool for studying parasites’ genetic variation, evolution and phylogenetic analysis. In order to control or eradicate human sparganosis it is necessary to establish a surveillance system that collects epidemiological data based on a fast, sensitive and reliable methodology to achieve unambiguous species identification of *Spirometra* tapeworms [14]. 

In Cuba, data on the presence of this organism are scarce. In 1947, Kourí reported for the first time that 20–30% of Cuban frogs *Osteopilus septentrionalis* species were infected [15]. Since then, only three cases have been reported in humans [16,17,18]. There are no further data on *Spirometra* infection in any other species of Cuban fauna, including domestic vertebrates.

In this article, we report three new cases of *S. decipiens* infection observed in two intermediate hosts and one definitive host. All samples were collected in San Manuel, a semi-rural town of Las Tunas province in the eastern region of the country.

## 2. Materials and Methods

### 2.1. Samples

A specimen of *Chilabothrus angulifer*, a species of endemic ophidian of Cuba, was captured in San Manuel, Las Tunas on July 2010 in a place characterized by a superficial slow fresh water course flowing across the savannah with some arboreal vegetation in the river bank. Twelve plerocercoids were obtained and fixed in 10% formalin from this snake. Eight years later, another endemic ophidian species, *Cubophis cantherigerus,* was captured 3 Km away from the site where the first snake was caught, and in a similar environmental condition. From this ophidian, 15 plerocercoids were sampled and fixed in 75% ethylic alcohol. Both ophidians were sacrificed for their fat, to be used for medicinal purposes. Finally, in 2011, one year after the first finding, one adult tapeworm was obtained from a juvenile domestic pet dog owned by a farmer family, in the same area where *Chilabothrus angulifer* was found. Figure 1 shows the field site where all samples were collected.

### 2.2. Histopathological Findings

All specimens obtained from both snakes and the dog were examined at the Pathology Department at the Institute of Tropical Medicine “Pedro Kourí”, Havana, Cuba by routine histological method using hematoxylin–eosin staining.

### 2.3. DNA Extraction

*Spirometra* genus was further corroborated using molecular biology techniques. DNA was extracted from the specimens obtained from each host (two ophidians and the dog) using the DNA mini kit for blood/tissue (QIAGEN, Hilden, Germany) according to the manufacturer’s instructions. All purified DNA samples were stored at −20 °C for further molecular analysis.

### 2.4. PCR Assays

For identification of *S. erinaceieuropaei* and *S. decipiens* species, PCR amplification of coding regions for the cytochrome b (the *cob* gene) and cytochrome c oxidase subunit 1 (*cox1* gene) which produce bands of different sizes were selected (Table 1). [14]

Mixtures for PCR amplifications of both genes were prepared for 100 ng of genomic DNA of each species with 10 pmol of each primer (Table 1), 25 mM MgCl_2_, 10× buffer, 10 mM dNTP mix, and 2.0 unit *Taq* polymerase (HotStar, Qiagen, Germany) in a final volume of 50 µL. The amplification protocol comprised an initial denaturation step at 94 °C for 3 min and 35 cycles of 94 °C for 1 min (denaturation), 45 °C for 1 min (annealing) and 72 °C for 1 min (extension). Finally, an extension at 72 °C for 5 min was programmed to complete synthesis of the PCR product. Genomic DNA of *S. decipiens* and *S. erinaceieuropaei* were kindly donated by Prof. Keeseon Eom and were used as positive controls. Sterile water was utilized as negative control in each PCR run.

### 2.5. Amplicons Detection

PCR products were visualized under UV exposition in transilluminator equipment (Macrovue 2011, LKB, Sweden) after running 15 µL in 1.6% agarose gel staining with ethidium bromide. 

## 3. Results

When the integument of both snakes was removed, at the level of the interfascicular tissue of the intercostal muscles surrounding the abdominal cavity, several elongated, tapered and pseudo-segmented larval forms of approximately 40 mm long by 5 mm wide were observed. These forms were pearly white and endowed with great mobility (contraction and relaxation) (Figure 2A), with a discreet invagination at the anterior end (primitive scolex) representing the bothrium (Figure 2B,C).

Under the light microscope, the histological sections stained with hematoxylin and eosin showed a body wall that varied in thickness and was composed of a layer of an external microvilli tegumentary layer, smooth muscle cells structured in a double layer and a single layer of tegumental cells. In the parenchyma, we found bundles of longitudinal muscle fibers following an irregular fashion, mesenchymal fibers, excretory channels and calcareous corpuscles in a loose stroma. No reproductive organs were observed (Figure 2D). Taken all together, the anatomical location, the host species infected with the specimens and their morphological features, the larvae were identified as plerocercoid larvae (sparganum) of *Spirometra* spp.

A domestic dog expelled through the anus an elongated and segmented organism approximately 25 cm long by 1 cm wide, with rectangular terminal segments, wider than it was long (Figure 3A).

Histological observation identified an integument composed of an outer acellular layer and an inner (subtegumentary) cell layer was also observed. Immediately below the nuclei of the tegumentary cells, vitellogenic glands organized into follicles or acini, distributed in the lateral fields of the ring were verified. They were composed of typical vitellogenic cells in different stages of maturation, which settle in the third stratum and spread through the dorsal and ventral faces of the ring. A thick, ill-defined band of longitudinal muscle fibers lies beneath the vitellogenic glands dividing the parenchyma into the cortical and medullary layers.

The testicular follicles in the middle or medullary layer and the gravid uterus were observed. The uterus containing elongated or oval eggs occupied the central area. The eggs showed more or less pointed ends, a smooth cover and the presence of a central mass of vitellogenic cells, flanking the fertilized ovum (formative vitellus). Despite the absence of the scolex in the recovered specimens, we considered that the samples in question were related to an adult of *Spirometra* spp. An adequate interpretation of the histological sections of the gravid rings in the horizontal plane and the characteristic spiral configuration of the uterus observed allowed us to identify the genus *Spirometra* and to differentiate it from *Dibotriocephalus* (*Diphyllobothrium*). According to Marty and Neafie, the specimens were classified as *S. decipiens* [19].

From each of the three samples (described in Section 2.3 DNA extraction), *S. decipiens*-specific bands (540 bp and 644 bp) were amplified in all PCR assays containing mixtures of *S. decipiens*-specific primers (Se/Sd-1800F and Sd-2317R; Se/Sd-7955F and Sd-8567R). On the contrary, *S. erinaceieuropaei*-specific bands (239 bp and 401 bp) were not amplified using a mixture of *S. erinaceieuropaei*-specific primers (Se/Sd-1800F and Se-2018R; Se/Sd-7955F and Se-8356R).

Additionally, a field study was carried out in the referred locality where the *S. decipiens* specimens were found, aiming to obtain information about the conditions that could facilitate the transmission of this parasite. Favorable environmental conditions for closing the biological cycle and the circulation of this organism were evident. An intermittent river slowly flows through the area that supports a biocenosis in which populations of *Cyclops* spp., *Rana catesbeiana* and fresh water fish of the genus *Gambusia* were identified. In the vegetation on the riverbank some birds, rodents and snakes abounded. In between the superficial water course and the houses of the neighboring settlement, there is a banana plantation where numerous specimens of *Osteopilus septentrionalis* dwell. During their reproductive period, they move to the lentic water course to perform the amplexus and oviposition and some days later it is possible to observe tadpoles swimming freely.

## 4. Discussion

The genus *Sparganum* was first described by Diesing in 1854 [20]. It comprises a group of similar larvae (plerocercoids) of broad tapeworms. Manson recovered numerous spargana from the tissues of a 34-year-old Chinese man and sent the specimens to Cobbold, who reported them as a new species in 1883. [21] Additionally, Manson and others recognized that these were immature tapeworms and suspected that they were related to Diphyllobothrium. In 1907, Verdum suggested that until the adult stage was identified, these worms be given the provisional generic name *Sparganum* [17].

The first patient known to be infected with *S. proliferum* was a Japanese woman who, in 1904, presented with subcutaneous nodules in the thigh and trunk. Ijima found proliferating larval cestodes in this nodule and named them *Plerocercoides prolifer*. Based on observations in 1927 on a patient who applied poultices containing dismembered frogs to her eyes, Casaux established the transmission of sparganosis by direct penetration of plerocercoids. Many researchers confirmed these findings [17]. The word Spirometra was first used by Faust et al. in 1929 but the genus *Spirometra* was settled in 1937 by Mueller [19].

The genus *Spirometra* belongs to the family Diphyllobothriidae and order Diphyllobothriidea. At first glance, *Spirometra* looks like *Dibotriocephalus* (formerly *Diphyllobotrium latum*) [5]. However, there are details documenting that they are different organisms. For example, the size of the evolutionary stages of the parasite is cited, being the adult worms (rarely exceeding 60 cm to 1 m) and the smallest proglottids in *Spirometra* sp. but the plerocercoid is much larger than *Dibotriocephalus*. One of the most important differential features is undoubtedly the morphology of the uterus, which is alluding to its generic name, spiral in *Spirometra* and rosette-shaped in *Dibotriocephalus*. Other differential morphological details are related to the location of the testes, the vitellogens, and the genital orifices [5,19].

Hyeong-Kyu Jeon and collaborators [22] noted that the main morphological characteristics for species differentiation from *Spirometra* spp. are the number of coils of their uterus: *S. erinaceieuropaei* (5–7 coils), *S. decipiens* (4–4½ coils) and *S. ranarum* (3 coils). 

Studies carried out based on morphology alone have distinguished around 50 nominal species of *Spirometra* [19,23]. Generally, species identification of *Spirometra* tapeworm is based on the anatomical description of adult worms; however, the typical specimens obtained in practice usually are *Spirometra* larval forms. It is hard to distinguish the morphological differences in the plerocercoids among spirometric tapeworms. Therefore, more species identification has preferred DNA barcoding methods. In this sense, the contribution of molecular studies to the diagnosis of this parasitosis is of paramount importance. 

Recently, differential identification of *Spirometra* species that infect humans can be established using a multiplex PCR technique [14]. In this assay, genomic amplification using four sets of primers allows for identifying spirometric parasites based on the length of the PCR products. The use of a specific PCR assay allowed us to identify *S. decipiens* from the three analyzed DNA samples. On the other hand, this PCR may be used for the differential identification of eggs, larvae and adult worms. At the same time, it can help trace back the source of intermediate hosts´ plerocercoid infection and identification of new intermediate hosts and final hosts, as well.

Reptiles are intermediate hosts of spargana and may contribute to the spread of the disease. When there is a high prevalence, consumption of reptile-infected raw meat poses a risk of sparganosis to humans. In some countries of Asia, snakes have a heavy infection of *Spirometra* spp. For instance, in Korea, around 50% of the reported cases of the disease are caused by the consumption of raw snake meat [23]. A similar situation was found in two out of the three cases reported in this work. 

*S. decipiens* has been identified in Brazil, China, and Korea [23]. Until a few years ago, in North America, only the *S. mansonoides* species was identified in cats and snakes (*Natrix natrix*) in New York State and Florida, respectively. Recently, *S. decipiens* was also identified in the United States of America. The authors of the report speculated that the spread of *S. decipiens* was caused by pets (e.g., dogs and cats) previously infected in South America [11].

In this study, the report of a domestic species (e.g., canids) involved in the biological cycle of this parasite is very important. The dog probably became infected when ingesting amphibians or snakes harboring plerocercoids. 

The characteristics of a fresh water ecosystem in this rural area—savannas with intermittent streams and surface water courses—contribute to the transmission of the *Spirometra* parasite. The importance of conducting environmental parasitology studies in this region in order to determine the impact of *Spirometra* spp. transmission in the local fauna and human health is because ecosystems such as this can be extensively found through the geography of the island. Nevertheless, no human case has been reported to date in the region under study.

In summary, it is very important to highlight that in the location referred to, the primary and secondary ecological factors necessary for the circulation of this organism converge. Future work based on the genomic sequence of *S. decipiens* will provide deeper knowledge about the ecological and behavioral factors that may drive the spread of this species in Cuba. 

## 5. Conclusions

To our knowledge, this paper is the first report of *S. decipiens* in species of Cuban endemic fauna and indeed in the Caribbean. These species constitute a real or potential risk of transmission of *Spirometra* species to humans in Cuba.

## Figures and Tables

**Figure 1 pathogens-11-01468-f001:**
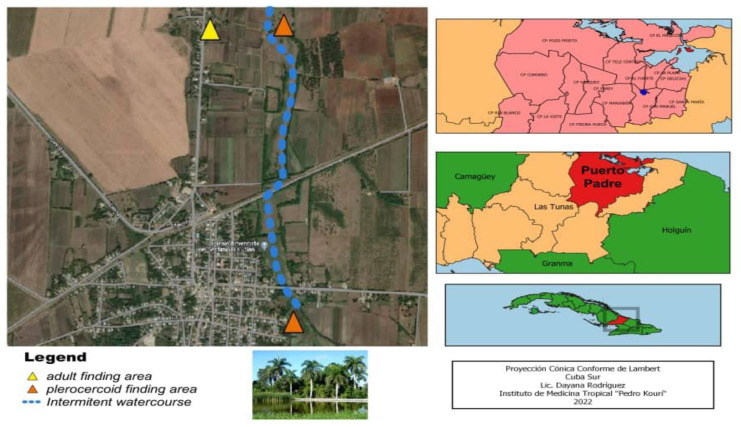
Geographic locality of all samples analyzed in this study.

**Figure 2 pathogens-11-01468-f002:**
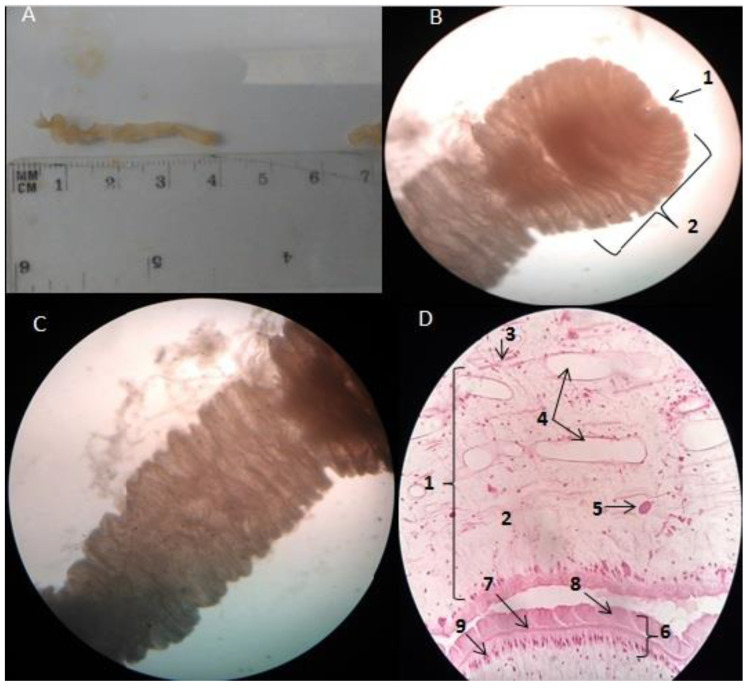
Plerocercoid larvae of *Spirometra* spp. (**A**) Gross appearance, 40 mm length × 5 mm width. (**B**) Anterior end: 1 bothrium, 2 primitive scolexes (40X). (**C**) Pseudo-segmented body (40X). (**D**) Microphotograph of a section of the parasite 1. Parenchyma, 2. Loose stroma, 3. Mesenchymal fiber, 4. Excretory channels, 5. Calcareous corpuscle, 6. Body wall, 7. Longitudinal muscle fibers, 8. Tegument, 9. Row of tegumentary cells (400X).

**Figure 3 pathogens-11-01468-f003:**
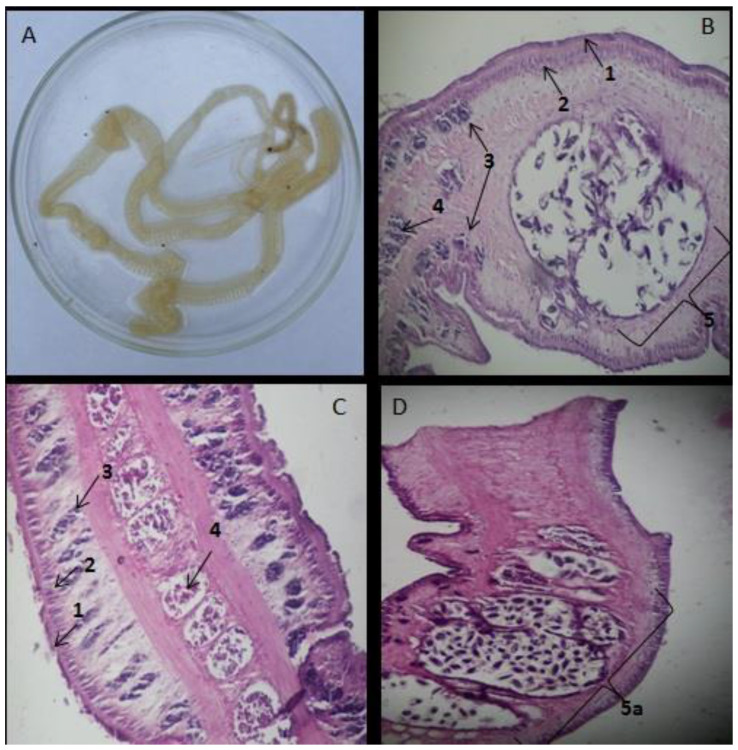
(**A**) Gross picture of an adult *Spirometra* spp., (**B**) Cross section of a gravid proglottid containing the central field, (**C**) Parasagittal section containing lateral field: 1. Acellular tegumentary layer, 2. Cellular tegumentary layer, 3. Vitellogenic glands, 4. Testes; (**D**) Horizontal section with: 5a. Spirally coiled uterus with four and a half convolutions. (Microphotographs **B**–**D** stained with HE/100×).

**Table 1 pathogens-11-01468-t001:** Nucleotide sequences of primers used for *Cob* and *Cox 1* gene amplification and identification of *Spirometra species*.

*Spirometra* Species Analyzed in This Study	Species-Specific Primers Gene *Cob* (5′-3′)	Amplicon Detection Gene *Cob* (bp)	Species-Specific Primers Gene *Cox 1* (5′-3′)	Amplicon Detection Gene *Cox 1* (bp)
*S. decipiens*	ID:24145860 Forward Se/Sd-1800F (AGTTATTTTCGGTTGGTGCTGTAG) Reverse Sd-2317R (TCCTCCCCCCACACGACAAAA)	540	ID:24145870 Forward Se/Sd-7955F (ACGTGGTTTGTGGTGGCTCATTTT) Reverse Sd-8567R (TTATTAACTTCCTAACCAACTTGATAC)	644
*S. erinaceieuropaei*	ID: 6446563 Forward Se/Sd-1800F (AGTTATTTTCGGTTGGTGCTGTAG) Reverse Se2018R (CCACAAACCCAATAACAAACTA)	239	ID: 6446566 Forward Se/Sd-7955F (ACGTGGTTTGTGGTGGCTCATTTT) Reverse Se-8356R (ATGATAGGGTATAGGTGACCA)	401

Se: *S. erinaceieuropaei*. Sd: *S. decipiens*.

## Data Availability

Not applicable.

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
