# Peer review of "Genetic and Morphological Identification of Spirometra decipiens in Snakes and Domestic Dog Found in Cuba"

_pathogens, 2022, doi:10.3390/pathogens11121468_

Round 1

Reviewer 2 Report

The study present the new data on the occurrence of the cestode Spirometra decipiens from Cuba, which is important from the medical and veterinary point of view.

I am of an opinion that the article fits into scope of Pathogens and could be published after major corrections – requires some additions regarding the methods used.

1. Abstract: It makes no sense to use the name "Spirometra spp." when a specific species of tapeworm has been identified. The abbreviation "spp." suggests that there were more than two species. Therefore, the abstract needs to be changed - rewritten.

Besides, the sentence " These specimens were  examined at the Pathology Department at the Institute of Tropical Medicine "Pedro Kourí " needs to be removed - this information adds nothing, especially to the abstract.

2. Please provide for each species (hosts, parasites) a complete name with author(s) and dates.

3. Line 47: should be Cyclopidae.

4. lines 47-48: How "wild animals" differ from amphibians and reptiles; whether amphibians and reptiles are not animals ? What do the Authors mean when they write "chicken" - is this a taxon/species ? This is not correct biological terminology - it needs to be corrected!!

In addition to common names (e.g., pigs), it is mandatory to use scientific names, since it is not clear what species are involved.

 5. line 83: „majá de Santa María” – please use the English names!

 6. line 91: The authors write that they used a routine histological study. However, these are not routine staining methods used for tapeworms. This is certainly why the images do not show the appropriate diagnostic features for this species (Figure 1, 2 and lines 144-162). For tapeworms, carmine staining is routinely used, then dehydration (e.g., alcohol series) is applied, followed by clearing and possibly mounting in Canada balsam. Therefore, I suggest that you supplement your work with preparations according to the method given or similar method. 

6. lines 136-142: It is understood that the larval stages of tapeworms have few diagnostic features, but this description (lines 136-142) really facilitated the determination to the genus Spirometra ?, whether these are diagnostic features

7. Figures 1 and 2: since the species was designated then the species name should be used and not "Spirometra spp.". The abbreviation "spp." as I mentioned, suggests that there were at least two species !

8. Please in the Materials and Methods chapter, or Results specify how many plerocercoids and adult stages of tapeworms were found; once it says "several" (line 131), and then (line 166) it is "in each three case”.

8. Please format the bibliography according to the Microorganisms MDPI.

Reviewer 3 Report

Overall, the manuscript is well organized and written (only some typing errors). The methods are well described. I have minor comments that should be addressed before the manuscript being appropriate for publication. Please consider specific comments as follows;

- Line 27: add full stop after Spirometra spp

- Line 36: Scientific name should be in italics.

- Line 83: Check spelling: endemic boidophidian

- Figure 1: Check unit 45 mm length x 0.5 mm width. Is this correct?

- Line 148: 3rd stratum, should be superscript.

- Line 152: add full stop after Spirometra spp

- Line 212: add full stop after Spirometra spp

- Line 251: add full stop after Spirometra spp

Materials and Methods

- 2.1 samples: Add more details of how specimens (sparganum and adult) were obtained and preserved (70%EtOH, 10%formaline, etc….).

- 2.4 PCR assays: Add more detail of the negative and positive control in PCR reactions.

Molecular result

- Is there any sequencing result? How many of samples that has sequenced?

Figures

- The scale bar should be added in the microscopic image.

References

The original of references should be added e.g.

- Line 184: Zhong et al. 1596, Line 187:Manson, The original paper of these authors should be cited instead of using the paper that cite the original.

- Paragraph 4 of introduction (Line58-66): The original paper of each species location should be added.

Round 2

Reviewer 2 Report

A few more comments on the scientific names.

1. I don't think I was understood correctly. Please give whole names (author and date) only for species (e.g. Spirometra decipiens); it is superfluous to give authors and dates for higher taxa (e.g. Amphibia, Reptilia, Rana etc., etc.) - this is generally not done, sometimes in typically taxonomic journals; here it is unnecessary.

2. When using a species name for the first time, please provide the common and scientific name, e.g. domestic cats Felis catus Linaneus, 1758; pig Sus scrofa Linnaeus, 1758. Then please use only one name, either the common name (domestic cat) or the scientific name F. catus.

3. I draw your attention to the correctness of the notation of scientific names - not always the author and the date are in bracket !!!, e.g. should be Sus scrofa Linnaeus, 1758. This type of notation is regulated by the appropriate nomenclature code, in this case the International Code of Zoological Nomenclature. Sometimes authors with the date are in bracket, and sometimes not !!! These designations mean something, for example, if the original species was placed in another genus - then the authors with the date are in bracket.

 4. Lines: 50-51: “Canids (Canis, Linnaeus 1758) and felines (Felidae G Fischer, 1817)” – should be “Dogs Canis and cats Felidae ….”. Please use the actual names.

5. Line 56-57: “…in cats (Felis silvestris catus Schreber, 1775), dogs (Canis lupus familiaris Linneaus, 1758)” – I know this may look strange, but the correct notation is: “…in domestic cats Felis catus Linaneus, 1758, domestic dogs Canis lupus familiaris Linneaus, 1758”. Besides, if we are talking about the domestic cat then its scientific name is different. Please check the veracity of all species names (scientific and common), because there are a lot of errors – e.g. in the Integrated Taxonomic Information System database.

 6. Locality and location: The term locality (e.g. line 100)  refers to a geographic locale of the external environment where the  parasites is found, and location refers to a habitat; please change.

  6. Scientific names – genus, species should be italicized, e.g. line 214.

7. It is worth explaining that sparganum is the larva – e.g. Discussion (lines 214-216.)
